# The Dissolution Behavior of Pyrite and Chalcopyrite in Their Mixture During Low-Temperature Pressure Oxidation: A Kinetic Analysis

**DOI:** 10.3390/ma18030551

**Published:** 2025-01-25

**Authors:** Kirill Karimov, Maksim Tretiak, Denis Rogozhnikov

**Affiliations:** Laboratory of Advanced Technologies in Non-Ferrous and Ferrous Metals Raw Materials Processing, Institute of New Materials and Technologies, Ural Federal University, Yekaterinburg 620002, Russia; kirill_karimov07@mail.ru (K.K.); darogozhnikov@yandex.ru (D.R.)

**Keywords:** pressure leaching, pyrite, chalcopyrite, sulfuric acid, pressure oxidation, sulfide, elemental sulfur, electrochemical couple, kinetic analysis

## Abstract

The research presented in this paper focused on the pressure leaching of pyrite and chalcopyrite in their mixture at a low temperature (100 ± 2 °C). The mathematical models of chalcopyrite and pyrite dissolution in their mixture are obtained. According to kinetic analyses, the oxidation process of chalcopyrite and pyrite is limited by intra-diffusion limitations. An elemental sulfur film passivates the surface of chalcopyrite and pyrite particles according to the SEM and EDX mappings. The data show that the oxidation mechanism of chalcopyrite and pyrite in their mixture has changed. The activation energy values of chalcopyrite and pyrite have increased from 51.2 to 59.0 kJ/mol, respectively. The oxidation degree of pyrite in its mixture with chalcopyrite increased significantly from 54.5 to 80.3% within 0–230 min. Copper and iron ions during oxidation were not associated with an increase in the dissolution degree of pyrite with the addition of chalcopyrite. The positive effect of pyrite in its mixture with chalcopyrite on its oxidation degree can be explained by the formation of an electrochemical bond between the minerals. Microphotographs and EDX mapping confirm that the positive effect of the chalcopyrite additive is correlated with a decrease in the formation of elemental sulfur on the pyrite surface. With no formation of conglomerates, the mineral’s sulfur content becomes more uniform, confirming the sulfides’ interaction with each other.

## 1. Introduction

In hydrometallurgical processes, one of the current trends is to perform leaching and kinetics at low temperatures, either in pressure reactors or under atmospheric pressure [1,2,3,4,5]. This approach has significant advantages, primarily because it reduces the consumption of reagents and energy. For instance, sulfur can be oxidized to its elemental form instead of sulfate, leading to reduced expenses for reagents and energy resources. Additionally, operating at lower temperatures reduces the need for expensive equipment. However, a challenge arises when elemental sulfur forms during these processes. The presence of sulfur can result in the formation of a passivating layer on the mineral surface, thereby significantly reducing the rate of reaction.

There are many ways to improve the leaching process, both under atmospheric and pressure conditions. These methods include mechanical activation, the use of surfactants (predominantly sodium lignosulfonate), and the addition of catalysts (Cu^2+^, Fe^3+^, and others) [6,7,8]. These techniques help break up the sulfide matrix, reduce the formation of passivating films, or eliminate them altogether. As a result, the efficiency of mineral opening is improved, leading to faster and more complete mineral oxidation during subsequent leaching with sulfuric, nitric, or mixed acid solutions [9,10,11,12,13,14,15].

The addition of catalysts significantly enhances the leaching processes of sulfide minerals by accelerating reaction rates or facility. These methods enhance the initiation of chemical reactions. Recent studies have highlighted the role of catalytic acceleration in leaching, particularly through oxidation-reduction mechanisms. For instance, research has demonstrated that iron ions serve as efficient catalysts during the acid leaching of uranium when oxidizing agents are present [16].

Studies on the oxidation of sulfide minerals in the presence of Fe (III) ions (10 g/L) [17] revealed a significant effect of temperature on the efficiency of oxidation of copper–zinc raw materials. The authors of Refs [17,18] established the catalytic effect of Ag during the leaching of chalcopyrite with a solution of iron (III) sulfate, which made it possible to reduce the leaching time from 2 h to 7 min and increase copper extraction into solution. The unfavorable kinetics of the reaction between chalcopyrite and the oxidizer, iron (III), were due to inhibition at the stage of electron transfer through the layer of elemental sulfur. When a soluble silver salt is added, the rate of the process increases significantly, and the reaction proceeds according to the following scheme [18] (Equation (1)):CuFeS_2_(s) + 4Ag^+^(aq) = 2Ag_2_S(s) + Cu^2+^(aq) + Fe^2+^(aq).(1)

The introduction of Ag^+^ ions, Ag_2_S, or a silver-containing concentrate into the pulp significantly accelerates the leaching of copper from chalcopyrite or other sulfide ores [19,20]. It has been shown [21] that Ag_2_S is a catalyst in CuFeS_2_ decomposition and an inhibitor in FeS_2_ decomposition. Carranza et al. [22] describe a technology of hydrometallurgical processing of chalcopyrite concentrates based on two-stage leaching with solutions containing iron (III). In this case, sulfides of non-ferrous metals are mainly leached at the first stage, and the chalcopyrite component is leached at the second one in the presence of Ag (I) ions as a catalyst. Total copper recovery into solution was >96% with a leaching time of 20 h.

At atmospheric pressure, under 90 °C and no oxygen presence, the leaching process can be conducted using pyrite as a catalyst, as demonstrated in the Galvanox process. In this method, pyrite forms a galvanic couple with chalcopyrite, significantly enhancing copper recovery. Studies have shown that, with a 410 mV redox potential (relative to a silver chloride electrode), a temperature of 85 °C, initial concentrations of 15 g/L H_2_SO_4_ and 5 g/L Fe, a pulp density of 7.8%, and a 2:1 ratio of pyrite to chalcopyrite, up to 80% copper recovery can be achieved within two hours [23]. The mechanism is attributed to the formation of a galvanic couple between FeS_2_ (pyrite) and CuFeS_2_ (chalcopyrite), where pyrite provides additional surface area for the reduction of Fe^3+^ ions. This interaction accelerates the anodic dissolution of chalcopyrite. The effectiveness of pyrite as a catalyst is strongly dependent on its quality, especially its silver content, which plays a critical role in enhancing the catalytic effect [24,25].

Research has demonstrated [26] the presence and combination of Fe^3+^ ions and O_2_ enhances chalcopyrite leaching enormously. Specifically, the chalcopyrite electrode potential increases by 53%, while the dissolution current density increases by 55 times. The potential increases by an additional 14% and the dissolution current density grows by 2.6 and 2.2 times when 42% FeS2 is introduced. Enhancing the catalytic activity of pyrite, particularly through silver treatment, further accelerates the leaching process. This effect is attributed to the formation of argentoyarosite (2AgFe_3_(SO_4_)_2_(OH)_6_), as reported in the Galvanox process [27]. Further studies [28,29,30] demonstrate that this acceleration is due to a galvanic mechanism. Researchers have observed that small amounts of silver, when applied through silver-treated pyrite, can enhance copper leaching in iron (III) ions solutions under atmospheric conditions. The proposed mechanism suggests that silver bonds with released sulfur in the form of Ag_2_S, maintaining electrical contact between chalcopyrite and pyrite. This promotes electron transfer and accelerates Fe (III) reduction to Fe (II) in the sulfate solution. In contrast, in the absence of silver, a sulfur layer forms, interrupting electrical conductivity and reducing efficiency.

Hiroyoshi et al. [31] identified the critical role of redox potential in chalcopyrite leaching with sulfuric acid. They found that active chalcopyrite dissolution occurs when the redox potential is below 0.7 V, governed by the Fe (II)/Fe (III) ion ratio in the solution. Moreover, studies [32] indicate that leaching efficiency improves significantly when both Fe (III) and Fe (II) are added into the solution, when combined with oxygen bubbling.

A detailed leaching mechanism for chalcopyrite has been proposed for chalcopyrite in sulfuric acid solutions containing dissolved O_2_, Fe (III), and Cu (II) ions [33]. Chalcopyrite is initially reduced to Cu_2_S by iron ions in the presence of Cu (II). Subsequently, Cu_2_S undergoes oxidation to Cu (II) ions and elemental sulfur by dissolved O_2_ and/or Fe (III). Since Cu_2_S is more easily oxidized than chalcopyrite, this intermediate step makes copper extraction more efficient. It has been established [34] that adding iron (II) ions to an iron (III) solution increases copper extraction by 1.54 times, and three times more copper passes into a solution also containing Cu(II) ions compared to a solution containing only iron (III) ions.

Despite these advancements, a consensus on the mechanisms of catalytic action in the electrochemical dissolution of sulfide minerals remains to be established. Most studies have focused on chalcopyrite dissolution in iron sulfate solutions, especially under atmospheric conditions and in the presence of pyrite. However, the behavior of these minerals under elevated temperatures and pressure conditions remains poorly understood.

Current scientific research in the field of refractory sulfide ore processing focuses on improving hydrometallurgical processes. Due to their high efficiency and potential for reducing energy and material consumption, technologies for mild autoclave and atmospheric leaching are of scientific and practical interest.

The aim of this study is to obtain mathematical models of the dissolution of chalcopyrite and pyrite in their mixture depending on the effects of oxygen pressure, initial concentration of sulfuric acid, concentrations of iron (III) and copper (II) ions, and time duration. The kinetic analysis focuses on overcoming diffusion limitations by analyzing the dissolution kinetics of pyrite and chalcopyrite, modeling their oxidation processes, and investigating the role of electrochemical interactions in enhancing their reactivity within a mixed-mineral system.

## 2. Materials and Methods

### 2.1. Analyses and Equipment Characterization

The chemical analysis of the original minerals and the resulting solid dissolution products was carried out using an ARL Advant’X 4200 wave-dispersive spectrometer (Thermo Fisher Scientific Inc., Waltham, MA, USA). The phase analysis was performed on an XRD 7000 Maxima diffractometer (Shimadzu Corp., Tokyo, Japan).

The granulometric analysis was performed by laser diffraction on an Analysette 22 Nanotec Plus (FRITSCH GmbH, Idar-Oberstein, Germany).

The chemical analysis of the obtained solutions was performed by inductively coupled plasma mass spectrometry (ICP-MS) on an Elan 9000 (PerkinElmer Inc., Waltham, MA, USA).

Scanning electron microscopy (SEM) was performed using a JSM-6390LV microscope (JEOL Ltd., Tokyo, Japan) equipped with a module for energy-dispersive X-ray spectroscopy analysis (EDX).

Experimental data were processed to obtain regression equations and Pareto diagrams using the Statgraphics Centurion Software Version 18 (Statgraphics Technologies, Inc., The Plains, VA, USA).

### 2.2. Materials and Reagents

The main raw materials were natural sulfide minerals: chalcopyrite (Vorontsovskoye deposit, Sverdlovsk region, Russian Federation), and pyrite, obtained from the Berezovskoye deposit (Sverdlovsk region, Russian federation). Their X-ray diffraction patterns are shown in Figure 1. All the minerals used were crushed and sieved on laboratory sieves, and the working fraction with a size of 80% of the class ≤ 40 μm was collected after sieving. The granulometric composition of the minerals is shown in Figure 2. The chemical composition of the minerals used is presented in Table 1. All the other reagents used were of analytical purity.

All the reagents used in this work were of analytical purity. The solutions were prepared from Fe_2_(SO_4_)_3_·9H_2_O, H_2_SO_4_, and CuSO_4_·5H_2_O dissolved in deionized water.

### 2.3. Equipment and Experimental Procedure

The laboratory experiments on pressure leaching were carried out on a titanium reactor with a volume of 1.0 L (Parr Instrument, Moline, IL, USA), with the possibility of feeding, adjusting the oxygen flow rate using a flow meter (Bronkhorst EL-FLOW Prestige and Bronkhorst EL-PRESS Metal-Sealed pressure regulators (AK Ruurlo, The Netherlands)), and temperature control. Mixing was carried out using an overhead stirrer to ensure pulp homogeneity.

Before each experiment, a pulp was prepared from sulfide minerals (20 g) weighed on an analytical balance and a 600 cm^3^ solution containing H_2_SO_4_, Fe (III), and Cu (II) of specified concentrations. The reactor’s filling factor was 0.6. After loading the pulp, the reactor was sealed, the stirrer was started, and the pulp was heated up to the required temperature of 100 °C. The stirrer rotation speed was maintained at 800 rpm, ensuring a uniform pulp density. When the specified temperature was reached, the reaction gas (oxygen) was supplied and the beginning of the experiment was recorded. Upon the completion of the experiment, the oxygen supply was stopped, and the reactor was cooled down to 70 °C. The pulp was filtered, and the cake was washed and dried to constant weight. The samples for analysis were taken from liquid and solid products.

### 2.4. Calculation Method

The dissolution degree of sulfide minerals in the mixture was calculated using the following method:1.The mass of copper in the concentrate was calculated using Equation (2):(2)$$m_{\mathrm{Cu}(\mathrm{conc})}\;=\frac{{\%\mathrm{Cu}}_{\mathrm{conc}}\times \;m_{\mathrm{sample}}}{100}$$
where %Cu_conc_ is the percentage of copper in the concentrate, %; and m_sample_ the mass of the sample, g.

2.The mass of copper in the cake was calculated using Equation (3):
(3)$$m_{\mathrm{Cu}(\mathrm{cake})}\;=\frac{{\%\mathrm{Cu}}_{\mathrm{cake}}\times m_{\mathrm{cake}}}{100}$$
where %Cu_cake_ is the percentage of copper in the cake, %; and m_cake_ the mass of the cake, g.

3.The mass of iron in the concentrate was calculated using Equation (4):
(4)$$m_{\mathrm{Fe}(\mathrm{conc})}\;=\frac{{\%\mathrm{Fe}}_{\mathrm{ct}}\times m_{\mathrm{sample}}}{100}$$
where %Fe_ct_ is the percentage of iron in the pyrite concentrate, %; and m_sample_ the mass of the sample, g.

4.The mass of iron in the cake was calculated using Equation (5):
(5)$$m_{\mathrm{Fe}(\mathrm{cake})}\;=\frac{{\%\mathrm{Fe}}_{\mathrm{cake}}\times m_{\mathrm{cake}}}{100}$$
where %Fe_cake_ is the percentage of iron in the cake, %; and m_cake_ the mass of the cake, g.

5.The percentage of chalcopyrite oxidation from the concentrate was calculated using Equation (6):
(6)$$\varepsilon_{Cu}=100-\frac{{\%\mathrm{Cu}}_{\mathrm{cake}}\times m_{\mathrm{cake}}}{{\%\mathrm{Cu}}_{\mathrm{conc}}\times m_{\mathrm{sample}}}\times 100$$
where %Cu_conc_ is the percentage of copper in the concentrate, %; m_sample_ the mass of the sample, g; %Cu_cake_ the percentage of copper in the cake, %; and m_cake_ the mass of the cake, g.

6.The percentage of pyrite oxidation was calculated using Equation (7):
(7)$$\varepsilon_{Fe}=100-\frac{{\%\mathrm{Fe}}_{\mathrm{cake}}\times m_{\mathrm{cake}}}{{\%\mathrm{Fe}}_{\mathrm{ct}}\times m_{\mathrm{sample}}}\times 100$$
where %Fe_ct_ is the percentage of iron in the concentrate, %; m_sample_ the mass of the sample, g; %Fe_cake_ the percentage of iron in the cake, and %; m_cake_ the mass of the cake, g.

ε_Cu_ and ε_Fe_ from Equations (6) and (7), respectively, are technological quantities that referred to the yield of the process.

## 3. Results and Discussion

### 3.1. Assessment of the Influence of the Main Parameters on the Low-Temperature Pressure Oxidation of Pyrite and Chalcopyrite in Their Mixture

Studies have indicated that pyrite significantly enhances copper extraction from chalcopyrite under atmospheric conditions at 85 °C. A mathematical experimental design was applied using an orthogonal central-compositional second-order approach to investigate this catalytic effect. The oxygen pressure, initial sulfuric acid concentration, iron (III) ion concentration, copper (II) ion concentration, and leaching duration were evaluated in this method [35]. The experimental variables included the oxygen pressure (X_1_), sulfuric acid concentration (X_2_), iron (III) ion concentration (X_3_), copper (II) ion concentration (X_4_), and leaching duration (X_5_). The constants were a temperature of 100 °C, a liquid-to-solid ratio of 10:1, and a pyrite-to-chalcopyrite ratio of 1:1. The independent variables were analyzed at five levels, with central values set at X_1_ = 0.5 MPa, X_2_ = 45 g/L, X_3_ = 6 g/L, X_4_ = 4.5 g/L, and X_5_ = 140 min.

The experimental data were processed to obtain mathematical models and diagrams using Statgraphics software, Version 19. The evaluation of the impact of each variable on the oxidation of chalcopyrite and pyrite was conducted by employing software data and graphical optimization tools [36].

#### 3.1.1. Effect of Parameters on the Oxidation of Chalcopyrite Mixed with Pyrite

Chalcopyrite reacts with sulfuric acid according to the following reactions:2CuFeS_2_ + 2H_2_SO_4_ + 7O_2_ = 2CuSO_4_ + Fe_2_(SO_4_)_3_ + S^0^ + 2H_2_O, ∆G (100 °C) =−2204.168 kJ/mol(8)CuFeS_2_ + H_2_SO_4_ + 2,5O_2_ = CuSO_4_ + FeSO_4_ + S^0^ + H_2_O, ∆G (100 °C) = −809.056 kJ/mol(9)CuFeS_2_ + 2Fe_2_(SO_4_)_3_ = CuSO_4_ + 5FeSO_4_ + 2S^0^, ∆G (100 °C) = −70.752 kJ/mol(10)CuFeS_2_ + 4O_2_ = CuSO_4_ + FeSO_4_, ∆G (100 °C) = −1238.377 kJ/mol(11)4FeSO_4_ + O_2_(g) + 2H_2_SO_4_ = 2Fe_2_(SO_4_)_3_ + 2H_2_O, ∆G (100 °C) = −304.494 kJ/mol(12)

Based on the described reactions, the sulfide sulfur in chalcopyrite can undergo oxidation via different pathways. Oxygen plays a key role in these processes, oxidizing sulfide sulfur into either elemental sulfur or sulfate ions (Equations (8) and (9)). Additionally, iron (III) ions act as oxidizing agents, facilitating the transformation of sulfide sulfur into elemental sulfur (Equation (11)). During the process, iron (II) ions are re-oxidized by oxygen to regenerate iron (III) ions, sustaining the leaching cycle.

The dissolution of chalcopyrite in a mixture with pyrite, influenced by the oxygen pressure, initial sulfuric acid concentration, iron (III) and copper (II) ion concentrations, and reaction time, can be quantitatively modeled. The resulting mathematical model demonstrates the combined effects of these parameters on the dissolution process. The equation representing this model is as follows:CuFeS_2_ (%) = 43.05 + 0.41 × X_1_ – 0.01 × X_2_ – 2.80 × X_3_ + 2.40 × X_4_ + 0.0349925 × X_5_ – 0.30 × X_1_^2^ – 0.01 × X_2_^2^ + 0.04 × X_2_X_4_ + +0.01 × X_2_X_5_ + 0.16 × X_3_^2^ – 0.61 × X_4_^2^,(13)
where X_1_ is the amount of oxygen, X_2_ is the initial concentration of sulfuric acid, X_3_ is the initial Fe^3+^ concentration, X_4_ is the initial Cu^2+^ concentration, and X_5_ is the time duration of the experiment.

Comparing the mean square value of each coefficient in the equation with the estimated experimental error revealed its statistical significance. The analysis revealed that several coefficients of X_1_^2^, X_1_X_2_, X_1_X_3_, X_3_X_4_, X_5_^2^, X_4_X_5_, X_3_X_5_, X_1_X_5_, X_1_X_4_, and X_2_X_3_ were statistically insignificant. As a result, these terms were excluded from the final equation to improve its precision and interpretability.

Figure 3 illustrates the correlation between the observed values of chalcopyrite dissolution and those predicted by the refined model. The reliability of the proposed model (Equation (13)) was validated by a close alignment between the predicted and experimental results, demonstrating its accuracy and robustness.

Figure 4 predicted response surfaces for the degree of chalcopyrite dissolution based on the model. The surfaces show the influence of oxygen pressure, initial sulfuric acid concentration, iron (III) ion concentration, copper (II) ion concentration, and leaching duration.

The results of this study show that all the examined parameters within the selected range have a statistically significant impact on the dissolution of chalcopyrite in a pyrite-containing mixture.

As shown in Figure 4a, the influence of oxygen pressure on chalcopyrite dissolution within the studied range is minimal. An increase in the oxygen pressure from 0.2 to 0.8 MPa results in only a moderate improvement in the dissolution degree, rising from 53% to 56% after 240 min of oxidation.

Figure 4b illustrates that increasing the initial concentration of sulfuric acid has a positive effect on the dissolution process. As the sulfuric acid concentration is increased from 20 g/L to 70 g/L, the degree of dissolution increases from 49% to 59% over a reaction duration of 240 min.

A similar trend is observed for the initial concentration of copper (II) ions, as presented in Figure 4c. Increasing the copper (II) ion concentration from 1 to 3 g/L improves the dissolution degree from 51% to 54% over the same reaction period.

The concentration of iron (III) ions, on the other hand, has a negative impact on the dissolution process, as shown in Figure 4d. The dissolution degree declines from 60% to 56% after 240 min of oxidation when the iron ion concentration is increased from 2 g/L to 10 g/L.

Summarizing the obtained information, it can be concluded that the oxidation of chalcopyrite under given conditions with pyrite is positively influenced by the partial oxygen pressure and initial acid and copper (II) concentrations. The enhanced formation of elemental sulfur through the oxidation of sulfide sulfur may explain the inhibitory effect of higher iron (III) ion concentrations. This elemental sulfur may form a passivating layer on the chalcopyrite surface, preventing further dissolution [37]. The obtained mathematical model is reliable, which is confirmed by high coefficient of multiple determination R^2^ = 0.998.

#### 3.1.2. Effect of Parameters on the Oxidation of Pyrite Mixed with Chalcopyrite

The reaction of pyrite with sulfuric acid in the presence of oxygen and iron (III) ions may proceed through the subsequent reactions:2FeS_2_ + 2H_2_SO_4_ + O_2_ = 2FeSO_4_ + 4S^0^ + 2H_2_O, ∆G (100 °C) = −368.076 kJ/mol(14)FeS_2_ + 3.5O_2_ + H_2_O = FeSO_4_ + H_2_SO_4_, ∆G (100 °C) = −1075.317 kJ/mol(15)4FeS_2_ + Fe_2_(SO_4_)_3_ + 4H_2_SO_4_ = 6FeSO_4_ + 9S^0^ + 4H_2_O, ∆G (100 °C) = −192.382 kJ/mol(16)2FeS_2_ + 3O_2_(g) + 2H_2_SO_4_ = Fe_2_(SO_4_)_3_ + 3S^0^ + 2H_2_O, ∆G (100 °C) = −1435.262 kJ/mol(17)4FeSO_4_ + O_2_(g) + 2H_2_SO_4_ = 2Fe_2_(SO_4_)_3_ + 2H_2_O, ∆G (100 °C) = −304.494 kJ/mol(18)

The presented reactions demonstrate that the sulfide sulfur in pyrite can be oxidized by oxygen to yield elemental sulfur and sulfate ions, as described in Equations (14) and (15). Iron (III) ions may also serve as oxidizers, facilitating the conversion of sulfur sulfide into its elemental form (Equation (16)). Additionally, iron (II) ions interact with oxygen, regenerating iron (III) ions (Equation (18)).

The dissolution of pyrite in a chalcopyrite–pyrite mixture, influenced by factors such as the oxygen pressure, initial sulfuric acid concentration, iron (III) ion concentration, copper (II) ion concentration, and leaching duration, is described by the following mathematical model:FeS_2_ (%) = 40.10 − 0.68 × X_1_ + 0.20 × X_2_ + 2.59 × X_3_ + 0.98 × X_4_ + 0.02 × X_5_ + 0.01 × X_1_X_5_ − 0.01 × X_2_^2^ + 0.03 × X_2_X_4_ + +0.01 × X_2_X_5_ − 0.12 × X_3_^2^ − 0.29 × X_4_^2^ − 0.01 × X_5_^2^,(19)
where X_1_ is the amount of oxygen, X_2_ is the initial concentration of sulfuric acid, X_3_ is the initial Fe^3+^ concentration, X_4_ is the initial Cu^2+^ concentration, and X_5_ is the time duration of the experiment.

The mean square values were compared to the experimental error estimate to assess the statistical significance of the coefficients. Based on the analysis, several coefficients were considered statistically insignificant. These include X_1_^2^, X_1_X_2_, X_1_X_3_, X_1_X_4_, X_3_X_4_, X_2_X_3_, X_3_X_5_, and X_4_X_5_. To improve the accuracy of the model, these terms were excluded from the final equation to improve accuracy.

Figure 5 presents a comparison between the actual dissolution degree of pyrite and the values predicted by the model. Equation (19) indicates that the selected model is reliable because of the close alignment between the observed and predicted results.

Figure 6 presents the response surfaces generated by the predictive model for the pyrite dissolution degree. The response surfaces are presented in Figure 6. These results account for the varying oxygen pressure, initial sulfuric acid concentration, concentrations of iron (III) and copper (II) ions, and reaction duration. When changing one of the parameters, the others were fixed at the average value inside the range selected.

All the examined parameters have a statistically significant positive effect on the dissolution of pyrite when mixed with chalcopyrite.

Increasing the oxygen pressure from 0.2 to 0.8 MPa has the greatest impact on the dissolution of pyrite (Figure 6a). The dissolution degree rises from 62% at 0.2 MPa to 78% at 0.8 MPa during the 240 min low-temperature pressure oxidation process. This trend highlights the critical role of oxygen pressure in enhancing the dissolution process.

The initial sulfuric acid concentration also significantly influences the pyrite dissolution (Figure 6b). When the concentration is increased from 20 g/L to 70 g/L, the dissolution degree improves from 64% to 71% over the same reaction period. This enhancement may be attributed to sulfuric acid reducing the formation of iron hydroxides and oxides, which can coat pyrite particles and hinder reagent access.

The concentration of iron (III) ions has a moderate positive effect on pyrite oxidation throughout the process (Figure 6c). An increase in concentration from 2 g/L to 7 g/L enhances the degree of dissolution from 67% to 69% within a duration of 240 min. Further increasing the concentration to 10 g/L yields minimal additional benefits.

Copper (II) ions have a minor, but statistically significant, positive impact on pyrite dissolution (Figure 6d). Increasing the concentration from 1 g/L to 2 g/L raises the dissolution degree from 74.9% to 78.5% at 240 min. However, concentrations beyond 3 g/L have little additional impact.

Under optimal conditions (temperature = 100 °C, Po_2_ = 0.8 MPa, [H_2_SO_4_] = 50 g/L, [Cu^2+^] = 3 g/L, [Fe^3+^] = 7 g/L, and duration = 240 min), the oxidation degree of pyrite reaches 80.3%. In contrast, chalcopyrite dissolution under these conditions is significantly lower at 56.4%. This disparity highlights the elevated reactivity of pyrite within the selected parameters.

The received data allow us to conclude that the oxidation of pyrite under given conditions in the mixture is positively influenced by all the considered parameters. The obtained mathematical model is reliable, which is confirmed by the high coefficient of multiple determination R^2^ = 0.988. This leads us to make an inference about the higher dissolution of pyrite at low-temperature conditions in a mixture with chalcopyrite, where the presence of chalcopyrite in the system is likely to be a decisive factor.

### 3.2. Kinetic Analysis of Sulfide Mineral Dissolution

The dissolution of chalcopyrite and pyrite may involve the formation of insoluble substances, such as elemental sulfur, on the surfaces of particles. Such products can inhibit reagent diffusion and indicate a surface-controlled reaction. Given the relatively low sensitivity of chalcopyrite dissolution to oxygen pressure, surface reaction kinetics may govern the process.

To model these heterogeneous reaction kinetics, the shrinking core model was used. This model assumes that reactions occur primarily at the particle surface, with the reaction zone progressively moving inward. As the process progresses, the unreacted core of the particle decreases, leaving behind a product layer. SCM identifies whether the process is diffusion-controlled (through the liquid layer or product layer) or reaction-controlled (on the particle surface). The slowest step, representing the greatest resistance, is rate-limiting, and its intensification can improve the leaching efficiency.

To determine the reaction mode and calculate the kinetic characteristics of pyrite and chalcopyrite dissolution, the shrinking core model (SCM) was employed, as previously reported [38,39]. Table 2 shows the main equations describing the SCM stages.

#### 3.2.1. Kinetic Analysis of Chalcopyrite and Pyrite Dissolution During Low-Temperature Pressure Leaching Without Mixing

The influence of temperature on the low-temperature pressure oxidation of chalcopyrite and pyrite is illustrated in Figure 7. These results further clarify the dissolution behaviors of these sulfide minerals under identical conditions.

As depicted in Figure 7a, the temperature exerts a significant influence on the dissolution of chalcopyrite. At 110 °C, the oxidation degree reaches 59.2% after 230 min. Conversely, reducing the temperature to 60 °C results in a significant decrease, with a dissolution rate of only 20.2% within the same duration. Increasing the leaching time from 20 to 240 min contributes to an increase in the chalcopyrite dissolution degree, and the positive effect of temperature is preserved over the entire time range of oxidation.

Based on the data in Figure 7b, temperature significantly affects the process of pyrite dissolution over the entire studied time interval. For example, at a temperature of 110 °C, the pyrite oxidation degree reaches 57.9% after 230 min. A decrease in temperature down to 60 °C reduces the dissolution degree to 20.5% over the same period.

The effect of temperature remains throughout the leaching period, with increased reaction time enhancing the dissolution rate. The correlation coefficients obtained when modeling the process of chalcopyrite and pyrite leaching separately with sulfuric acid solutions are shown in Table 3.

As shown in Table 3, the correlation coefficients for the shrinking core model (SCM) equations indicate that Equation (1) provides the best fit for the experimental data. The dissolution processes of chalcopyrite and pyrite are governed by intra-diffusion, with reagent diffusion through the solid reaction product layer acting as the rate-limiting step.

The activation energy was calculated using the Arrhenius equation (Equation (20)). Plots of lnkc versus 1/T were constructed, where kc is the angular coefficient derived from Figure 8a,c. The coefficient calculated when constructing the straight line y = ax + b in these coordinates determines the slope of the curve. The activation energy (Ea) was determined by the slope of the linear relationship, yielding values of 51.2 kJ/mol for chalcopyrite and 50.6 kJ/mol for pyrite (Figure 8b,d).lnk_c_ = lnA − Ea/RT(20)

Based on the slope of the lines obtained by substituting into Equation (1), which describes the diffusion of reagents through the product layer, the empirical order regarding oxygen pressure was determined. The empirical partial orders for oxygen pressure were determined by plotting lnk_c_ against lnPo_2_ under oxygen pressures of 0.2–0.8 MPa at a temperature of 100 °C. As a result, an empirical partial order for oxygen pressure of 0.272 for pyrite and 0.075 for chalcopyrite, respectively, was obtained. These low empirical partial orders for oxygen pressure also confirm the internal diffusion limits during the oxidation processes of pyrite and chalcopyrite (Figure 9). Based on the empirical partial orders’ values, it can be inferred that oxygen pressure has a negligible impact on chalcopyrite dissolution, in contrast to pyrite. This fact may indicate the formation of a denser film of products on the surface of its particles, which could lead to more serious internal diffusion limitations.

Furthermore, when plotting graphs of the dependencies of ln kc vs. ln(Cu)/ln(Fe)/ln(H_2_SO_4_), empirical partial orders for the initial concentration of copper (II) and iron (III) ions and sulfuric acid were calculated for chalcopyrite and pyrite, where kc is the slope calculated similarly. For chalcopyrite, the partial orders were 0.14, 0.12, and −0.31 for copper (II), iron (III), and sulfuric acid, respectively. The negative effect of sulfuric acid concentration is attributed to the increased oxidation of sulfide sulfur to elemental sulfur one (Equation (8)), which passivates particle surfaces [34,38,40].

For pyrite, the partial orders were 0.13, 0.14, and 0.27 for copper (II) and iron (III) ions and sulfuric acid, respectively.

By integrating the Arrhenius equation into No. 1, Table 3, which models the diffusion of reagents through the product layer, the following generalized equation can be derived:1 − 3 × (1 − x)^2/3^ + 2 × (1 − x) = k_o_ × Po_2_^n^ × C_Cu_^n^ × C_Fe_^n^ × C_H2SO4_^n^e^−Ea/RT^ × t,(21)
where *n* is the empirical partial order for the component.

Based on the experimental data, the generalized kinetic equations for the oxidation of chalcopyrite and pyrite at a low-temperature pressure are formulated as follows (Equations (22) and (23), respectively):1 − 3 × (1 − x)^2/3^ + 2 × (1 − x) = k_o_ × Po_2_^0.075^ × C_Cu_^0.136^ × C_Fe_^0.119^ × C_H2SO4_^−0.313^ × e^−51188/RT^ × t(22)1 − 3 × (1 − x)^2/3^ + 2 × (1 − x) = k_o_ × Po_2_^0.272^ × C_Cu_^0.128^ × C_Fe_^0.137^ × C_H2SO4_^0.265^ × e^−50553/RT^ × t(23)

Subsequent graphical analyses of all experimental conditions (temperatures, initial concentrations of copper (II) and iron (III) ions and sulfuric acid) enabled the estimation of a fixed slope for chalcopyrite and pyrite, calculated as *a* = 0.134 × 10^5^ and a = 0.142 × 10^5^, respectively. These values correspond to the pre-exponential factor (k_0_) and were confirmed by linear fitting with high correlation coefficients (R^2^), as depicted in Figure 10.

According to the data obtained from Figure 10, the general kinetic equations for chalcopyrite and pyrite have the following form (Equations (24) and (25), respectively):1 − 3 × (1 − x)^2/3^ + 2 × (1 − x) = 0.134 × 10^5^ × Po_2_^0.075^ × C_Cu_^0.136^ × C_Fe_^0.119^ × C_H2SO4_^−0.313^ × e^−51188/RT^ × t(24)1 − 3 × (1 − x)^2/3^ + 2 × (1 − x) = 0.142 × 10^5^ × Po_2_^0.272^ × C_Cu_^0.128^ × C_Fe_^0.137^ × C_H2SO4_^0.265^ × e^−50553/RT^ × t(25)

According to the data provided, the oxidation process of chalcopyrite and pyrite proceeds with intra-diffusion limitations. The rate is principally restricted by the diffusion of reagents through the solid reaction product layer. According to Equations (8), (15) and (16), an elemental sulfur film on particle surfaces significantly inhibits reagent access to the reaction zone during low-temperature pressure oxidation [32,34,40]. Furthermore, the low empirical partial orders for oxygen pressure reinforce the existence of intra-diffusion limitations during the oxidation process. The negative effect of increasing the sulfuric acid concentration on chalcopyrite oxidation can be explained by the increased conversion of the sulfide sulfur into an elemental one (Equation (9)) and the screening of the surface of its particles [34].

Separate kinetics calculations of mineral oxidation without their mixing allow us to conclude that the reactions are limited by internal diffusion through the product layer. This conclusion is confirmed by the values of activation energy and reaction orders for oxygen, copper, and iron ions and sulfuric acid.

Further kinetic calculations of the oxidation of minerals at their mixing in a proportion of 1:1 were carried out.

#### 3.2.2. Kinetic Analysis of Chalcopyrite and Pyrite Dissolution During Low-Temperature Pressure Leaching in a 1:1 Mixture

The impact of temperature on the low-temperature pressure oxidation of chalcopyrite and pyrite in a 1:1 mixture is illustrated in Figure 11.

As shown in Figure 11a, the temperature significantly influences the dissolution of chalcopyrite. At a temperature of 110 °C, up to 50.6% of the chalcopyrite is oxidized in 230 min of the process. Conversely, reducing the temperature to 60 °C results in a lower dissolution degree of 17.4% over the same duration.

An increase in the leaching time from 20 min to 240 min consistently increases the chalcopyrite dissolution degree. This trend persists across all the temperatures, demonstrating that temperature maintains its positive effect throughout the oxidation process.

Figure 11b shows that temperature has a significant impact on the dissolution of pyrite during the time studied. For example, at a temperature of 110 °C, the pyrite oxidation degree reaches 78.5% after 230 min. Reducing the temperature down to 60 °C reduces the dissolution degree down to 23.6% over the same period.

Like chalcopyrite, extending the leaching time from 20 to 240 min results in a significant increase in the degree of pyrite dissolution. However, pyrite oxidation appears to be more sensitive to temperature changes, with higher temperatures increasing the dissolution rates at all the time intervals.

The dissolution kinetics were modeled using the shrinking core model (SCM), with the correlation coefficients (R^2^) presented in Table 4. No. 1 best describes the experimental data, with the highest correlation coefficient. This indicates that both chalcopyrite and pyrite oxidation occur under internal diffusion control, where reagent diffusion through the solid reaction product layer limits the process.

The activation energy was calculated by plotting the dependence of the natural logarithm of the slope kc on the inverse temperature T, similar to the previous calculations. The apparent activation energy was determined graphically (Figure 12). The calculated activation energies were 59.0 kJ/mol for chalcopyrite and 74.6 kJ/mol for pyrite, confirming that pyrite oxidation is more temperature-dependent than chalcopyrite oxidation.

Figure 13 shows the results of calculating the empirical partial orders for oxygen pressure. The following values were obtained: 0.330 for pyrite and 0.059 for chalcopyrite, respectively. These low empirical partial orders for oxygen pressure confirm the intra-diffusion limitations for pyrite and chalcopyrite oxidation (Figure 13).

The empirical partial orders suggest that oxygen pressure has minimal influence on the dissolution of chalcopyrite, unlike pyrite. This difference may be caused by the formation of a denser product layer on the surface of chalcopyrite particles. This layer may act as a barrier, increasing intra-diffusion limitations and limiting the access of reagents to the reaction surface.

The empirical partial orders for the low-temperature pressure leaching of chalcopyrite and pyrite in a 1:1 mixture were determined based on the initial concentrations of copper (II) ions, iron (III) ions, and sulfuric acid. The partial orders determined for chalcopyrite were 0.30 for copper (II) ions, 0.17 for iron (III) ions, and 0.36 for sulfuric acid. Unlike chalcopyrite monosulfide oxidation, an increase in the concentration of iron (III) ions in a mixture with pyrite has a detrimental impact. This negative impact is likely caused by the oxidation of the sulfide sulfur to an elemental one, which forms a passivating layer on the chalcopyrite surface, reducing reagent access and hindering dissolution [37].

For pyrite, the partial orders were 0.13 for the copper (II) concentration, 0.14 for the iron (III) concentration, and 0.27 for sulfuric acid. Unlike chalcopyrite, the positive values suggest that increasing the concentration of all three reagents enhances the dissolution of pyrite.

To derive generalized kinetic equations, the coefficients ko were determined graphically, similarly as described earlier. The graphical results and corresponding correlation coefficients (R^2^) are displayed in Figure 14.

According to the data in Figure 14, the general kinetic equations for chalcopyrite and pyrite have the following form (Equations (26) and (27), respectively):1 − 3 × (1 − x)^2/3^ + 2 × (1 − x) = 0.231 × 10^5^ × Po_2_^0.059^ × C_Cu_^0.300^ × C_Fe_^−0.178^ × C_H2SO4_^0.356^ × e^−59002/RT^ × t(26)1 − 3 × (1 − x)^2/3^ + 2 × (1 − x) = 0.133 × 10^5^ × Po_2_^0.330^ × C_Cu_^0.141^ × C_Fe_^0.108^ × C_H2SO4_^0.218^ × e^−74580/RT^ × t(27)

According to the data provided, the oxidation process of chalcopyrite and pyrite in a 1:1 mixture proceeds with internal diffusion limitations. The process is limited by the diffusion of reagents through the solid reaction product layer. During low-temperature pressure oxidation, the particle surfaces of chalcopyrite and pyrite are passivated by elemental sulfur films, as described in Equations (8) and (14) [40].

The initial sulfuric acid concentration significantly impacts chalcopyrite dissolution. For chalcopyrite alone, the empirical partial order is negative (−0.313). Nevertheless, the addition of pyrite in a ratio of 1:1 results in a positive value (+0.356), indicating a shift in the oxidation mechanism and interaction between the two minerals present in the mixture.

For chalcopyrite monosulfide, enhancing the initial concentration of iron (III) ions had a favorable effect on its dissolution, with an empirical partial order of 0.12. However, when pyrite was added in a 1:1 ratio, this trend reversed. The empirical partial order changed to −0.18, likely due to the increased oxidation of sulfide sulfur to elemental sulfur. This reaction forms a barrier layer on the chalcopyrite surface, reducing the accessibility of reagents [34].

The initial concentrations of copper, iron, and sulfuric acid showed a consistent positive response to pyrite dissolution. This trend was observed in both pure pyrite and in mixtures with chalcopyrite, highlighting the robust reactivity of pyrite in these conditions.

The analysis of the data indicates that the oxidation mechanism of chalcopyrite and pyrite in their mixture has changed. The activation energy of chalcopyrite increased from 51.2 to 59.0 kJ/mol, whereas it increased for pyrite from 50.6 to 74.6 kJ/mol. This suggests that the mixed oxidation process has a stronger temperature dependence.

This effect is especially noticeable for pyrite, namely: with the addition of chalcopyrite, the activation energy increases by 24 kJ/mol and reaches a value of 74.6 kJ/mol. During the initial period of the mineral oxidation process, the greatest positive effect of temperature was observed during the first 30 min (Figure 7 and Figure 11). The change in the nature of the curves for pyrite was more pronounced. For chalcopyrite, the addition of pyrite had a noticeable effect only in the first 30 min of dissolution. Subsequently, the graphs approach a plateau, whereas during the oxidation of an individual sulfide, the degree of dissolution increased throughout the process. The overall dissolution degree of chalcopyrite decreases across the entire temperature range in a 1:1 mixture with pyrite within 230 min.

In order to make the discussion more transparent, the comparison results of activation energies and reaction orders for chalcopyrite and pyrite oxidation reactivities both without mixing and in a 1:1 mixture are given in Table 5.

### 3.3. Analysis of the Cakes from Low-Temperature Pressure Dissolution of Chalcopyrite and Pyrite

Based on the kinetic data, the leaching process is limited by internal diffusion, which is probably related to the screening of the sulfide surface by elemental sulfur. To confirm these conclusions, analyses by X-ray phase spectrometry and scanning electron microscopy were carried out.

The X-ray diffraction patterns of oxidation cakes from the low-temperature pressure dissolution of chalcopyrite, pyrite, and their 1:1 mixture are shown in Figure 15.

The data presented in Figure 15 indicate that all the cakes contain under-oxidized minerals and elemental sulfur. Its content in the cakes is 24.8% for chalcopyrite, 11.3% for pyrite, and 19.2% for their mixture. The intensity of the pyrite peaks in the X-ray diffraction pattern for the mixed-mineral cake is lower than that of the chalcopyrite peaks, which suggests that pyrite is more reactive in the mixture.

Micrographs and EDS mappings of oxidation cakes for chalcopyrite, pyrite, and their 1:1 mixture are shown in Figure 16. These were obtained under identical conditions as the X-ray analysis.

According to the data presented in Figure 16a,d, after low-temperature pressure leaching, chalcopyrite particles are characterized by both smooth and loose, non-uniform surfaces. The elemental distribution is highlighted by the red zones for sulfur, green zones for iron, and yellow zones for copper in the EDS mapping (Figure 16d,g,j,m). Chalcopyrite is represented by a combination of these zones. The presence of elemental sulfur is also noticeable as bright red growths on the chalcopyrite surface (Figure 16d). The distribution of sulfur on the chalcopyrite surface is non-uniform and leads to the formation of conglomerates.

Pyrite particles also exhibit a mixture of smooth and irregular surfaces, as shown in Figure 16b,e. Small sulfur conglomerates have been observed on the surfaces of rectangular particles. EDS mapping (Figure 16e,h,k) confirms the presence of sulfur (red zones) and iron (green zones), representing pyrite. Bright red sulfur growths prominently cover pyrite surfaces, especially on developed or rough areas (Figure 16e). Sulfur is the predominant component in smaller particles, whereas sulfur is distributed unevenly on larger particles with smooth surfaces.

Figure 16c,f illustrate the presence of two distinct types of particles within the oxidized cake. The first type, pyrite, exhibits smooth surfaces interspersed with pits and caverns due to the oxidation reaction (Figure 16c,f). Chalcopyrite, a second type, has irregular surfaces with noticeable defects and inclusions of various shapes (Figure 16c,f). Unlike the dissolution behavior observed for individual minerals, particle conglomerates are not present on the surface during the leaching of pyrite and chalcopyrite mixtures.

The distribution of the elemental sulfur depends on the oxidation process. Sulfur distribution is more uniform across the particles in mixtures of pyrite and chalcopyrite. The sulfur (red zones), iron (green zones), and copper (yellow zones) distributions are illustrated in Figure 16i,l,n, respectively.

Figure 17 presents micrographs of cakes obtained following the low-temperature pressure oxidation of individual minerals and their 1:1 mixtures, as well as the compositional analysis at specific points. The elemental compositions corresponding to these micrographs are detailed in Table 6.

When oxidizing chalcopyrite not in a mixture, elemental sulfur conglomerates were also found on its surface (Figure 17a,d). The total content of elemental sulfur on the surface is 7.4–73.6% (Figure 17c point 2,3 and Figure 17f point 4). When oxidizing chalcopyrite in its mixture with pyrite, its surface has no conglomerates, but has pronounced defects and inclusions of various shapes. Elemental sulfur is also evenly distributed over all the particles, and its total content decreases up to 5.1–9.4% (Figure 17c point 2,3 and Figure 17f point 4).

The obtained data show that during the low-temperature pressure leaching of pyrite not in a mixture, elemental sulfur conglomerates are formed on the surface of the particles. The total content of elemental sulfur on its surface is 2.3–72.6% (Figure 17b,e). When pyrite is oxidized in its mixture with chalcopyrite, its surface contains no conglomerates. Elemental sulfur is distributed evenly over all the particles and its total content decreases down to 0.1–1.9% (Figure 17c point 1,4 and Figure 17f point 1–3).

The increased dissolution degree of pyrite in the presence of chalcopyrite is not driven by an increase in the concentration of copper (II) or iron (III) ions, according to kinetic data from the low-temperature pressure oxidation of chalcopyrite and pyrite. These ions had a negligible impact on the degree of dissolution, both for the individual minerals and their mixtures. The obtained empirical partial orders confirm this; their values almost do not change during oxidation in a mixture or separately.

According to data in the literature, pyrite may form galvanic bonds with other sulfides exhibiting semiconductor properties. These bonds serve as an alternative reactive surface for oxidation reactions through electrochemical contact. Importantly, pyrite oxidation in such systems occurs with minimal formation of elemental sulfur, particularly during the initial stages of the reaction [41].

The observed improvement in pyrite oxidation within its mixture with chalcopyrite is attributed to the formation of an electrochemical bond between the two minerals. Other researchers have widely documented this interaction. In the presence of strong oxidants such as oxygen or iron (III) ions, the minerals form an electrochemical pair. The electron transfer between the minerals through the passivating film limits the dissolution process [42]. Unlike the Galvanox™ process, where galvanic interactions predominantly enhance chalcopyrite oxidation, low-temperature pressure oxidation conditions allow for the oxidation of both pyrite and chalcopyrite. In this context, pyrite acts as a source of catalytic interaction, facilitating its own dissolution and that of chalcopyrite under the influence of electrochemical pairing [2].

This theory is confirmed by the change in the oxidation mechanism of chalcopyrite and pyrite in their mixture. This is indicated by the increase in the activation energy values during oxidation of minerals in the mixture: from 51.2 up to 59.0 kJ/mol for chalcopyrite and from 50.6 up to 74.6 kJ/mol for pyrite. This means that the effect of temperature increases during the oxidation of minerals in the mixture, which is observed in the initial period of the process, during the first 30 min (Figure 7 and Figure 11). In this case, the change in the nature of the curves is more pronounced for pyrite. For chalcopyrite, the addition of pyrite has a positive effect in the first 30 min of dissolution only, and then it has a negative effect, since the curves almost reach a plateau, whereas during the oxidation of individual sulfides, the dissolution degree increases during the entire duration. The total oxidation degree of chalcopyrite in the mixture decreases for the entire temperature range over 230 min.

Changes in the influences of initial concentrations of sulfuric acid and iron (III) ions in the mixture are further evidence of the formation of an electrochemical bond between pyrite and chalcopyrite, which follows from the obtained values of empirical partial orders.

Microphotographs and EDX mapping confirm that the positive effect of the chalcopyrite additive is associated with a decrease in elemental sulfur formation on the pyrite surface. The elemental sulfur content decreases up to 5.1–9.4% on the chalcopyrite surface, while for pyrite its content decreases up to 0.1–1.9%. The elemental sulfur distribution on minerals becomes more uniform with no formation of conglomerates, which also confirms their interaction with each other.

In the initial period, pyrite and chalcopyrite oxidation occurs at the maximum rate. However, the dissolution of chalcopyrite slows over time due to surface passivation caused by the elemental sulfur. The oxidation of pyrite also decreases during this period, though less sharply than when chalcopyrite is absent. The gradual passivation of pyrite surfaces with elemental sulfur explains this reduced rate.

The observed reduction in sulfur accumulation and improved distribution in mixed systems highlight the benefits of chalcopyrite addition for pyrite oxidation. This interaction mitigates surface passivation and sustains higher oxidation rates, particularly in the initial stages of the process. Such findings are pivotal for optimizing hydrometallurgical methods, where managing elemental sulfur is critical for maintaining efficient mineral dissolution.

These findings demonstrate the enormous significance of electrochemical interactions in mixed sulfide oxidation systems. By leveraging these dynamics, hydrometallurgical processes can be optimized to achieve improved dissolution efficiency, even under less aggressive temperature and pressure conditions.

## 4. Conclusions

This study explored the low-temperature pressure oxidation of pyrite–chalcopyrite, both not mixed and in a mixture, and developed a mathematical model describing their dissolution behavior. The model incorporates variables such as oxygen pressure, initial sulfuric acid concentration, the concentration of iron (III) ions and copper (II) ions, and duration.

The results of the experiments revealed that increasing the oxygen pressure from 0.2 to 0.8 MPa had a minimal effect on the dissolution of chalcopyrite. Increasing the initial sulfuric acid concentration from 20 to 70 g/L significantly enhanced chalcopyrite dissolution. The process was negatively impacted by increasing concentrations of iron (III) ions. The dissolution rate of pyrite grew from 62% to 78% over 240 min after the oxygen pressure was increased from 0.2 to 0.8 MPa. In addition, higher initial concentrations of sulfuric acid, copper (II), and iron (III) ions have also improved pyrite dissolution under these conditions.

It was found that the improved dissolution of pyrite in the sulfide mixture was not directly related to the increased concentrations of copper (II) and iron (III) ions. The empirical partial orders indicate that these ions had a minimal impact on the dissolution rates of both minerals, which remained virtually unchanged throughout the oxidation process of either the mixture or the individual minerals. Instead, it was attributed to the formation of an electrochemical interaction between the two minerals that enhanced the dissolution of pyrite in the mixture.

More evidence of mineral interaction was provided by microphotographs and EDX mapping, which indicated that the presence of chalcopyrite reduced elemental sulfur accumulation on the pyrite surface. The distribution of the elemental sulfur was more uniform, with no examples of conglomerate formation, indicating the electrochemical interaction between the minerals. The influence of sulfuric acid and iron (III) ions on chalcopyrite oxidation also changed when pyrite was added, which showed that the mixture had a modified oxidation mechanism.

Chalcopyrite addition for pyrite oxidation is highlighted by the observed reduction in sulfur accumulation and improved distribution in mixed systems. Higher oxidation rates are sustained in the initial stages of the process due to this interaction. These discoveries are crucial for enhancing hydrometallurgical techniques, where effectively managing elemental sulfur is imperative for ensuring efficient mineral dissolution.

Further studies are intended to provide a clearer understanding of the electrochemical link between these sulfide minerals and their interaction with elemental sulfur under low- and medium-temperature pressure conditions. The disclosure of the nature of the interaction between pyrite and chalcopyrite and the behavior of sulfur is one of the urgent tasks for hydrometallurgical processes and has a promising prospect of implementation.

## Figures and Tables

**Figure 1 materials-18-00551-f001:**
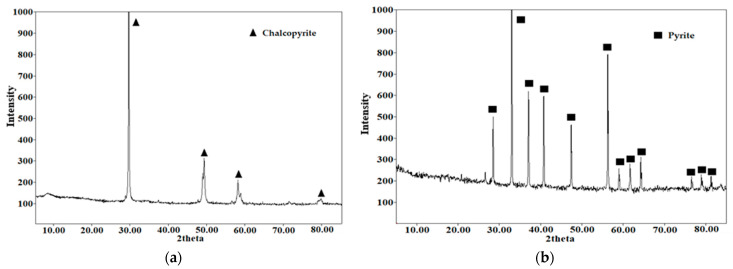
X-ray diffraction pattern of phase composition; (**a**)—chalcopyrite; and (**b**)—pyrite.

**Figure 2 materials-18-00551-f002:**
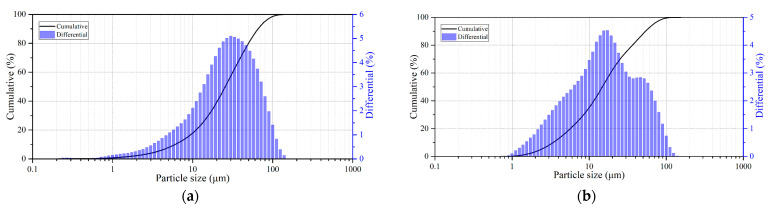
Particle size distribution of sulfide minerals; (**a**)—chalcopyrite; and (**b**)—pyrite.

**Figure 3 materials-18-00551-f003:**
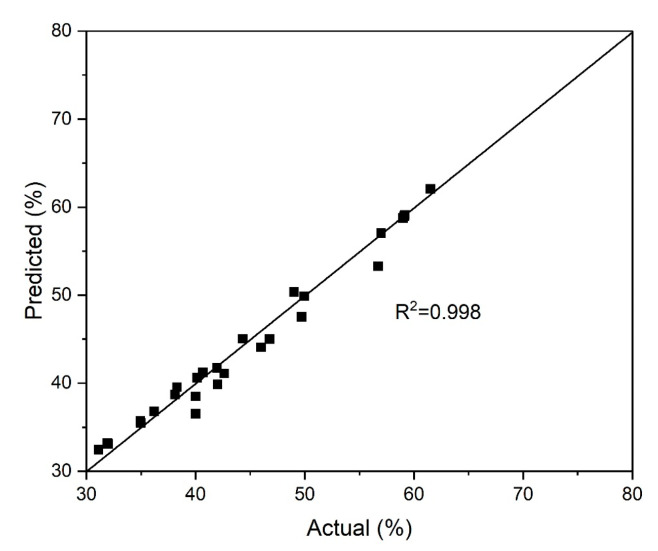
Comparison between actual and predicted values for the degree of chalcopyrite dissolution in a pyrite–chalcopyrite mixture.

**Figure 4 materials-18-00551-f004:**
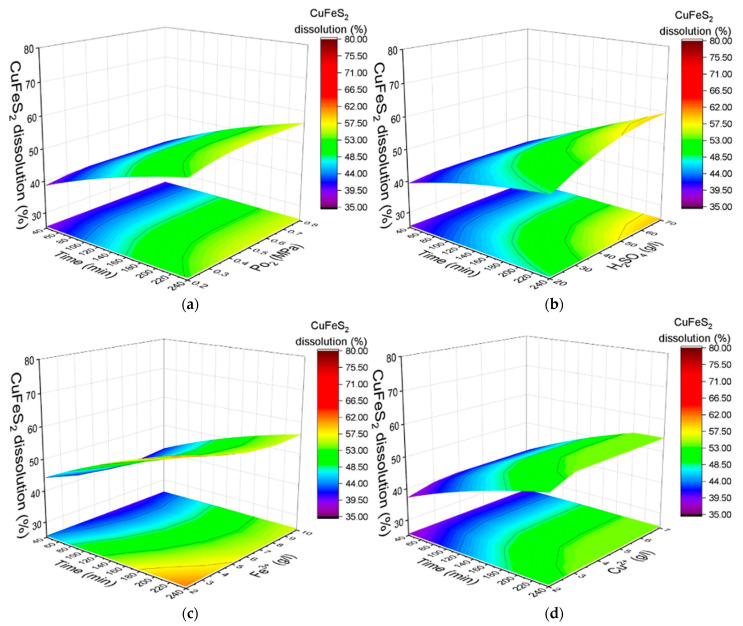
Dependence of the degree of chalcopyrite dissolution on various factors: (**a**) duration and oxygen pressure, (**b**) initial sulfuric acid concentration, (**c**) iron (III) ion concentration, and (**d**) copper (II) ion concentration.

**Figure 5 materials-18-00551-f005:**
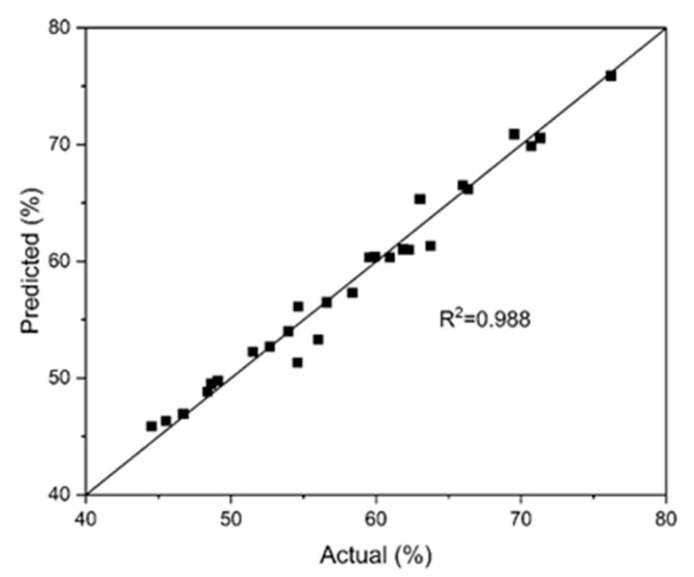
Comparison of actual and predicted values of pyrite dissolution in its mixture with chalcopyrite.

**Figure 6 materials-18-00551-f006:**
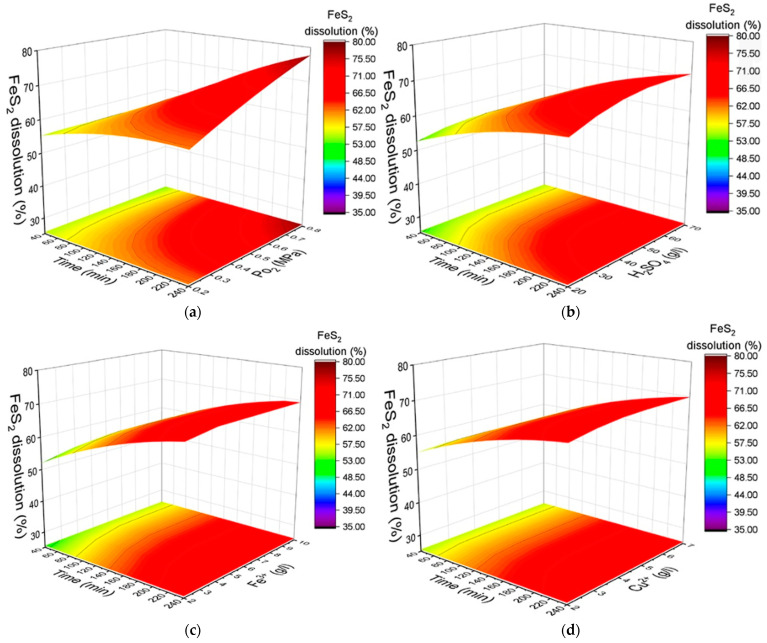
Response surface showing the dependence of pyrite dissolution on the (**a**) duration and oxygen pressure, (**b**) initial sulfuric acid concentration, (**c**) iron (III) ion concentration, and (**d**) copper (II) ion concentration.

**Figure 7 materials-18-00551-f007:**
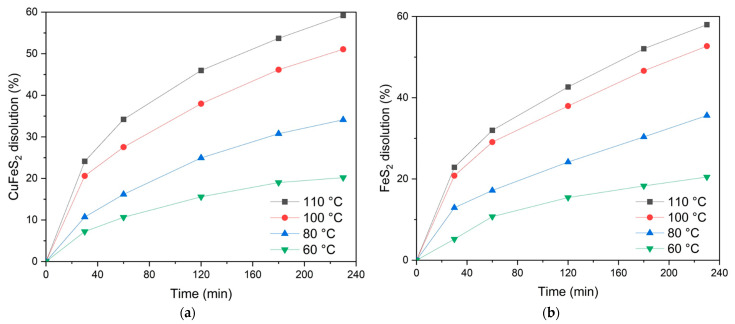
The dissolution degree of chalcopyrite (**a**) and pyrite (**b**) depends on temperature.

**Figure 8 materials-18-00551-f008:**
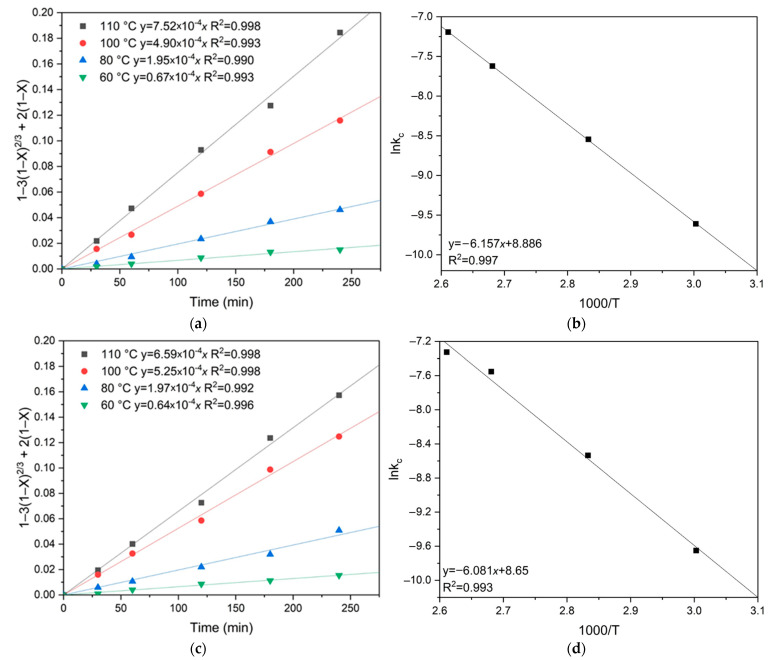
Calculation of the slope k_c_ for (**a**) chalcopyrite and (**c**) pyrite depending on temperature, and graphical determination of the activation energy for (**b**) chalcopyrite and (**d**) pyrite.

**Figure 9 materials-18-00551-f009:**
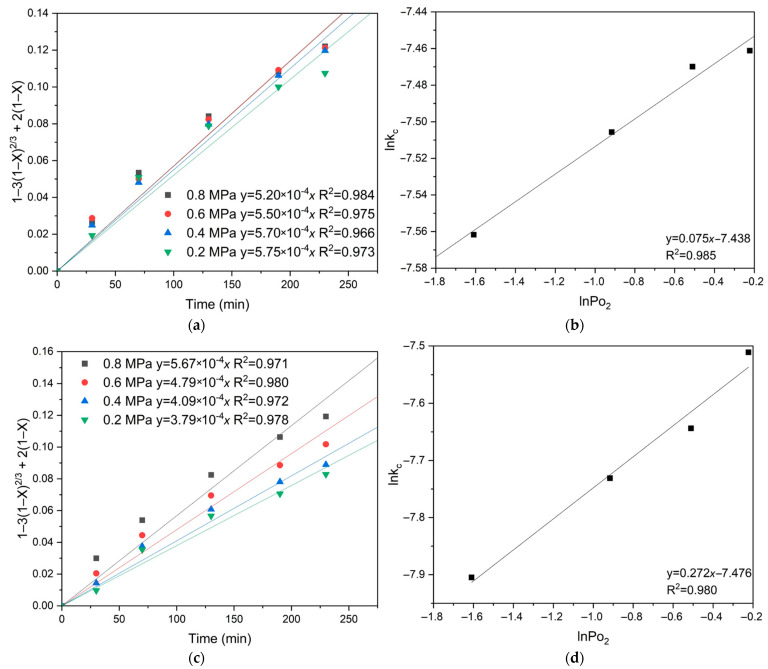
Calculation of the slope *k*_c_ for (**a**) chalcopyrite and (**c**) pyrite depending on oxygen pressure, and graphical determination of the empirical partial order for oxygen pressure for (**b**) chalcopyrite and (**d**) pyrite.

**Figure 10 materials-18-00551-f010:**
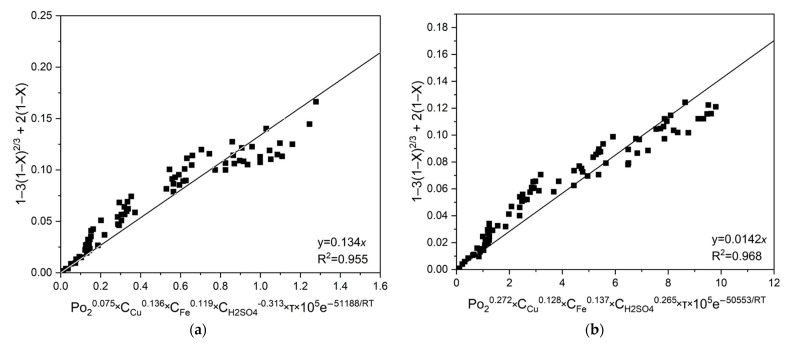
Graph to estimate *k*_0_ for (**a**) chalcopyrite and (**b**) pyrite.

**Figure 11 materials-18-00551-f011:**
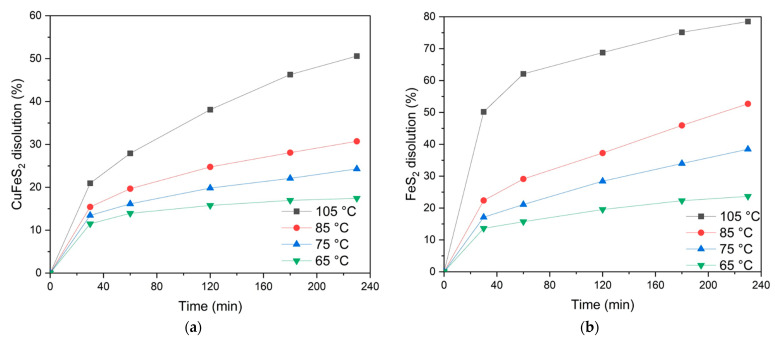
Temperature dependence of the dissolution degree of (**a**) chalcopyrite and (**b**) pyrite.

**Figure 12 materials-18-00551-f012:**
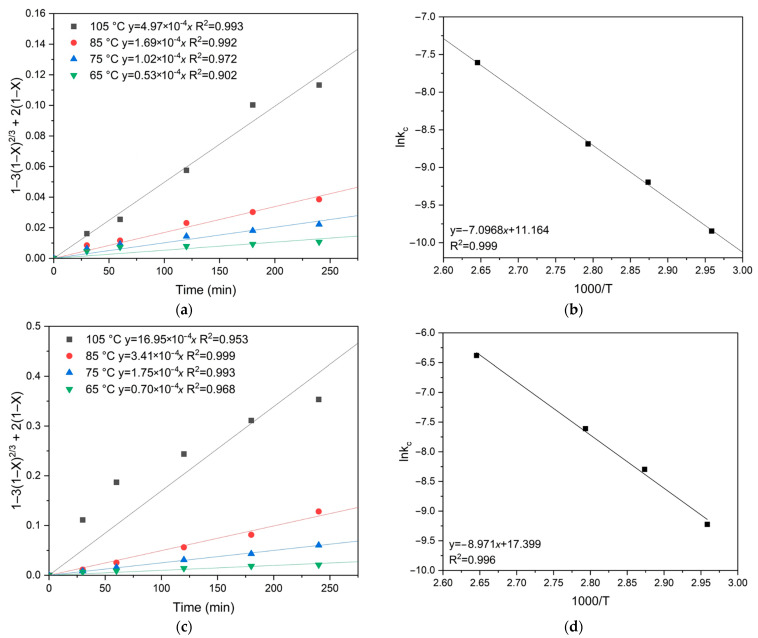
Calculation of the slope *k*_c_ for chalcopyrite (**a**) and pyrite (**c**) depending on temperature, and graphical estimation of activation energy for chalcopyrite (**b**) and pyrite (**d**).

**Figure 13 materials-18-00551-f013:**
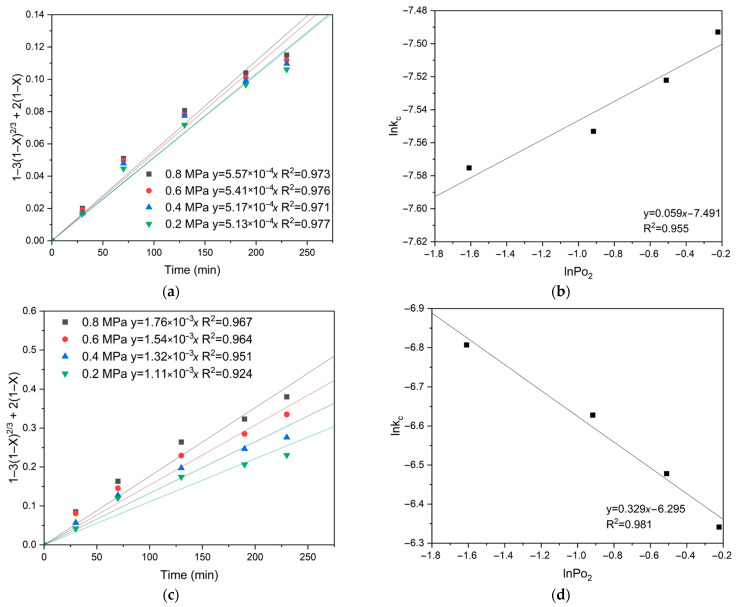
Calculation of the slope *k*_c_ for chalcopyrite (**a**) and pyrite (**c**) depending on the oxygen pressure, and graphical estimation of the empirical partial order for oxygen pressure for chalcopyrite (**b**) and pyrite (**d**).

**Figure 14 materials-18-00551-f014:**
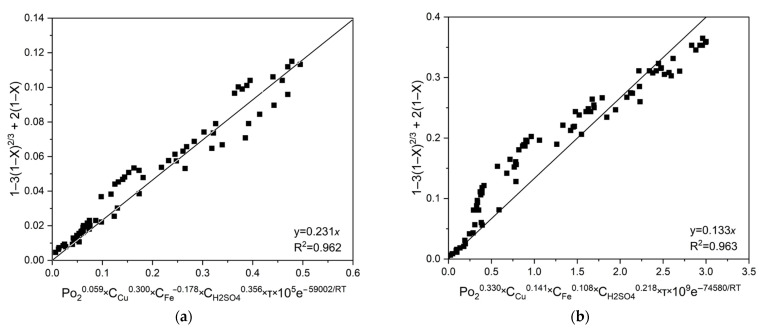
Graph to estimate *k*_0_ for chalcopyrite (**a**) and pyrite (**b**).

**Figure 15 materials-18-00551-f015:**
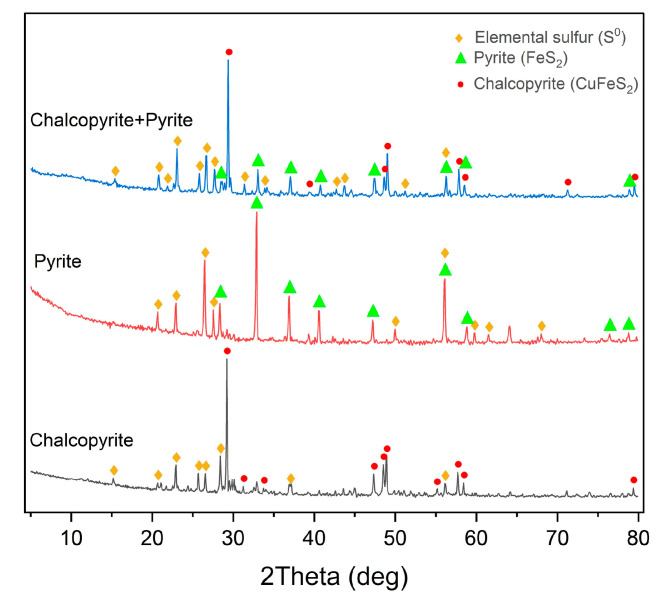
X-ray diffraction pattern of the cakes from low-temperature pressure oxidation of pyrite, chalcopyrite, and their mixture in a ratio of 1:1 (*t* = 100 °C, P_O2_ = 0.8 MPa, [H_2_SO_4_] = 50 g/L, [Cu^2+^] = 3 g/L, [Fe^3+^] = 10 g/L, and duration 230 min).

**Figure 16 materials-18-00551-f016:**
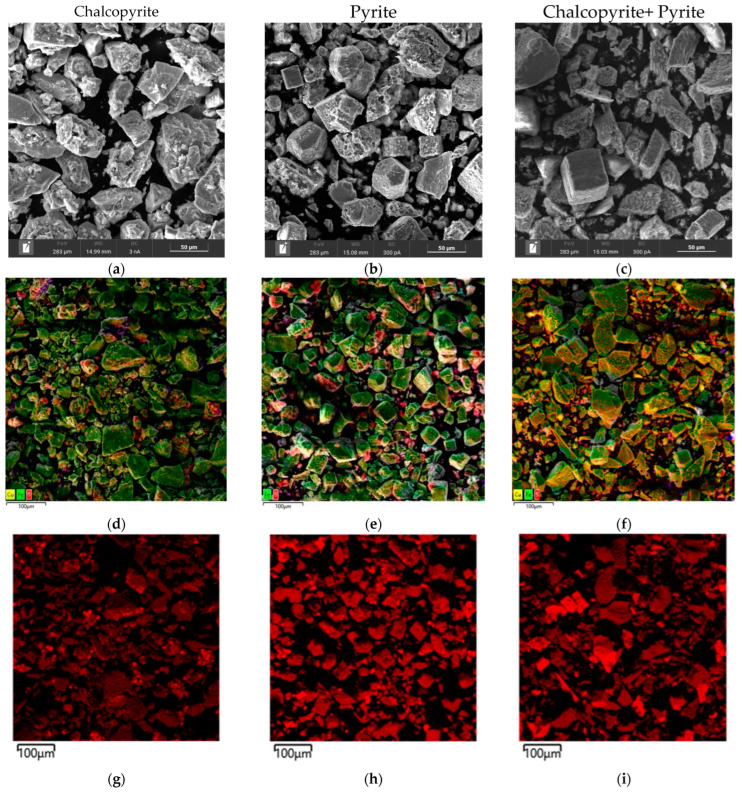
SEM images of oxidation cake particles (**a**–**c**) and combined EDS mapping (**d**–**f**) for sulfur (**g**–**i**), iron (**j**–**l**), and copper (**m**,**n**).

**Figure 17 materials-18-00551-f017:**
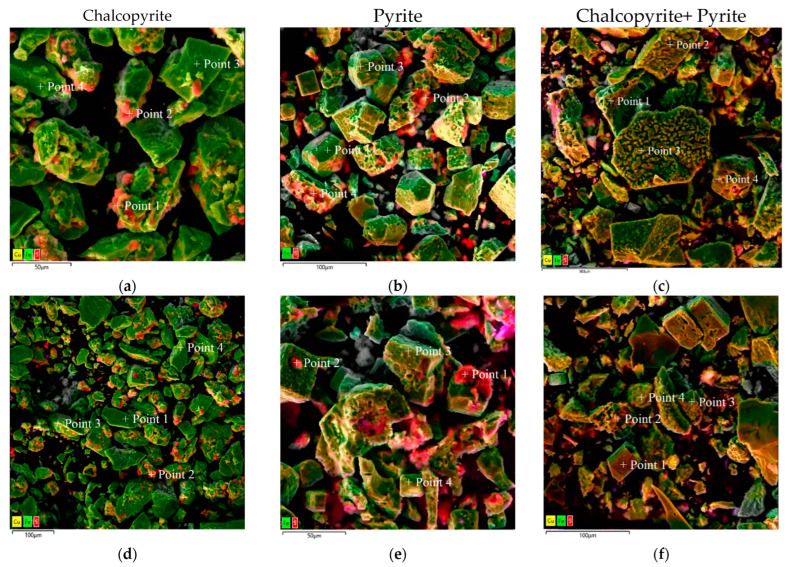
Micrographs of the cakes after low-temperature pressure oxidation of (**a**,**d**) chalcopyrite, (**b**,**e**) pyrite, and (**c**,**f**) their mixture with composition determination points.

**Table 1 materials-18-00551-t001:** The chemical composition of the minerals used.

Material	Content/wt.%			
	Cu	Fe	S	Others
CuFeS_2_	33.4	32.0	33.6	1.0
FeS_2_		44.1	50.8	5.1

**Table 2 materials-18-00551-t002:** Shrinking core model equations.

#	Limiting Step	Equation
1	Diffusion through the product layer (sp)	1 − 3(1 − X)^2/3^ + 2(1 − X)
2	Diffusion through the product layer (pp)	X^2^
3	Diffusion through the product layer (cp)	X + (1 − X)ln(1 − X)
4	Diffusion through the liquid film (sp)	X
5	Surface chemical reactions (cp)	1 − (1 − X)^1/2^
6	Surface chemical reactions (sp)	1 − (1 − X)^1/3^
7	Diffusion through the product layer, interfacial limiting step (new model)	1/3ln(1 − X) + [(1 − X)^−1/3^ − 1]

sp—spherical particles, pp—prismatic particles, and cp—conus particles.

**Table 3 materials-18-00551-t003:** Correlation coefficients of the SCM equations for modeling chalcopyrite and pyrite dissolution separately from each other.

#	Equation	R^2^			
		60 °C	80 °C	100 °C	110 °C
CuFeS_2_					
1	1 − 3(1 − X)^2/3^ + 2(1 − X)	0.993	0.990	0.993	0.998
2	X^2^	0.929	0.925	0.936	0.956
3	X + (1 − X)ln(1 − X)	0.948	0.945	0.953	0.949
4	X	0.932	0.939	0.928	0.941
5	1 − (1 − X)^1/2^	0.952	0.953	0.965	0.947
6	1 − (1 − X)^1/3^	0.958	0.938	0.968	0.938
FeS_2_					
1	1 − 3(1 − X)^2/3^ + 2(1 − X)	0.996	0.992	0.998	0.998
2	X^2^	0.930	0.915	0.936	0.961
3	X + (1 − X)ln(1 − X)	0.948	0.905	0.923	0.975
4	X	0.937	0.937	0.952	0.954
5	1 − (1 − X)^1/2^	0.948	0.950	0.942	0.945
6	1 − (1 − X)^1/3^	0.959	0.963	0.957	0.960

**Table 4 materials-18-00551-t004:** Correlation coefficients of the SCM equations for modeling chalcopyrite and pyrite dissolution in a 1:1 mixture.

#	Equation	R^2^			
		65 °C	75 °C	85 °C	105 °C
CuFeS_2_					
1	1 − 3(1 − X)^2/3^ + 2(1 − X)	0.902	0.972	0.992	0.993
2	X^2^	0.888	0.968	0.957	0.956
3	X + (1 − X)ln(1 − X)	0.841	0.941	0.971	0.964
4	X	0.827	0.911	0.883	0.938
5	1 − (1 − X)^1/2^	0.830	0.920	0.889	0.953
6	1 − (1 − X)^1/3^	0.957	0.891	0.923	0.831
FeS_2_					
1	1 − 3(1 − X)^2/3^ + 2(1 − X)	0.968	0.993	0.999	0.953
2	X^2^	0.923	0.933	0.916	0.919
3	X + (1 − X)ln(1 − X)	0.948	0.905	0.936	0.944
4	X	0.879	0.936	0.922	0.842
5	1 − (1 − X)^1/2^	0.886	0.969	0.921	0.876
6	1 − (1 − X)^1/3^	0.888	0.953	0.943	0.887

**Table 5 materials-18-00551-t005:** The comparison results of kinetics analysis.

Definition	Value			
	Not Mixed		1:1 Mixture	
	CuFeS_2_	FeS_2_	CuFeS_2_	FeS_2_
E_a_, kJ/mol	51.2	50.6	59.0	74.6
Oxygen partial order	0.075	0.272	0.059	0.330
Cu (II) partial order	0.14	0.13	0.3	0.14
Fe (III) partial order	0.12	0.14	-0.18	0.11
H_2_SO_4_ partial order	-0.31	0.27	0.36	0.22

**Table 6 materials-18-00551-t006:** Element contents at the composition determination points, %.

Element	Fe	Cu	S_sulfide_	S^0^	Other	Total
Figure 17a. Point 1	14.8	13.6	13.7	54.5	3.4	100.0
Figure 17a. Point 2	5.6	5.0	5.0	73.6	10.8	100.0
Figure 17a. Point 3	26.4	27.8	28.0	7.2	10.6	100.0
Figure 17a. Point 4	28.9	31.7	31.9	7.4	0.1	100.0
Figure 17d. Point 1	28.0	30.5	30.7	7.4	3.4	100.0
Figure 17d. Point 2	15.3	14.9	15.0	39.4	15.4	100.0
Figure 17d. Point 3	25.4	26.4	26.6	8.7	12.9	100.0
Figure 17d. Point 4	25.0	26.0	26.2	7.3	15.5	100.0
Figure 17b. Point 1	43.9	-	50.3	3.1	2.7	100.0
Figure 17b. Point 2	17.1	-	19.6	51.4	11.9	100.0
Figure 17b. Point 3	38.6	-	44.2	4.7	12.5	100.0
Figure 17b. Point 4	8.5	-	9.7	72.6	9.2	100.0
Figure 17e. Point 1	14.1	-	16.2	59.0	10.7	100.0
Figure 17e. Point 2	19.5	-	22.3	47.2	11.0	100.0
Figure 17e. Point 3	40.4	-	46.3	2.3	11.0	100.0
Figure 17e. Point 4	40.2	-	46.1	3.7	10.0	100.0
Figure 17c. Point 1	41.3	-	47.3	0.1	11.3	100.0
Figure 17c. Point 2	27.1	29.1	29.3	5.1	9.4	100.0
Figure 17c. Point 3	25.8	27.7	27.9	9.4	9.2	100.0
Figure 17c. Point 4	41.6	-	47.7	0.1	10.6	100.0
Figure 17f. Point 1	44.5	-	51.0	0.6	3.9	100.0
Figure 17f. Point 2	43.7	-	50.1	1.9	4.3	100.0
Figure 17f. Point 3	40.1	-	46.0	0.1	13.8	100.0
Figure 17f. Point 4	27.3	29.1	29.3	5.9	8.4	100.0

## Data Availability

The original contributions presented in the study are included in the article, further inquiries can be directed to the corresponding author.

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
