# Peer review of "The Dissolution Behavior of Pyrite and Chalcopyrite in Their Mixture During Low-Temperature Pressure Oxidation: A Kinetic Analysis"

_materials, 2025, doi:10.3390/ma18030551_

Round 1

Reviewer 1 Report

Comments and Suggestions for Authors

Dear Authors,

Your study is methodically and substantively supported. It is an interesting and well-executed research work. The analysis techniques used were purposeful and allowed for obtaining experimental results for statistical analysis.

Congratulations on the concept and execution.

Below I present my remarks/suggestions/comments, according to the chronoogy of reading the manuscript.

I recommend them to your attention.

- The entire manuscript should be checked in terms of the publisher's guidelines, the way of citing literature in the body of the study, the spacing between section and subsection titles, etc.

- The References section should be verified and corrected with particular care!

- The numbering of subsections in section 3 should be corrected

Section 1.

- It is recommended to cite researchers by name instead of a descriptive form, e.g.;

Japanese researchers [27] = Nazari et all [27]

Ref. [18] = Carranza et all [18]

line 65- equation should be provided according to the rules of chemistry

line 85- the wording "that the presence of Fe³⁺ ions and O₂ significantly" suggests the presence of O2 ions, should be rephrased to avoid ambiguity

Section I should be concluded with an indication of the research objectives presented in the manuscript - this should be supplemented

Section 2

- indexes (a), (b) for marking figures, are placed in the upper left corner of the graph

- if the axes of the graph are not described, they should be described, and if figures imported from the measuring device software are inserted, the relationships presented there should be explained - this should be supplemented

- editor's guidelines indicate that figures and graphs should have comparable dimensions - if possible

- in the explanations of the markings used (equations), the units and definition should be indicated -if necessary

-the parameters described by equations 5 and 6 are correct and correctly defined, please consider introducing or supplementing that they are technological quantities -referred to as yield

Section 3.

-equations 7-11 are correct, to increase their readability, please make a table - it will be easier to compare thermodynamic values ​​for individual reactions

-I recommend changing the resolution of figures 3 and 4

-please specify, supplement, explain in more detail, the information contained in lines 250-263; -what change in copper recovery would be sufficiently satisfactory (how many percent?), if a change in oxygen addition increasing the yield by 3% has a minimal impact on efficiency? can such a result for the tested parameter be neglected or omitted? and if so, why? Similar experimental results in lines 315-317 were defined as: "moderate impact"

-equations 13-17 are correct in terms of the quality of the reactants, however fractional values ​​of stoichiometric coefficients are not appropriate. They should be corrected.

To increase readability, please make a table - it will be easier to compare thermodynamic values ​​for individual reactions.

If the Authors want to indicate the quantitative (molar) proportions of reagents for which the thermodynamic value of the chemical reaction was determined, this should be included in the manuscript as comments on the results for the evaluation of the results of a given experiment, or the thermodynamic constant should be recalculated for the selected amount of the main reagents (pyrite, chalcopyrite) and placed in the table

- table 2, expanding the abbreviation in the title

- section 3.2.2,

I do not question the validity of the verification of the SCM kinetic model for leaching, but the results presented in figure 7 are arranged according to the first-order kinetic dependence for a typical chemical reaction.

Has such an approach been analyzed? It may be worth considering it and supplementing the calculations. I leave it to the Authors to decide.

The SCM model approximation of the results presented in Fig.9 is not perfect. The use of one model is methodologically justified, but the presented results may indicate a variable order of chemical reaction kinetics with the passage of time. I recommend it for future consideration.

- The results of analyses based on the used SCM model are statistically correct, due to the use of factor analysis. The determined coefficients of kinetic equations are correct, and the model-to-experiment fit index is high. Without a doubt, such an approach to assessing the significance of basic parameters on the results of the leaching process is an added value of this manuscript and proves the knowledge and analytical skills of the researchers. They also constitute an interesting description of the mechanisms responsible for chemical reactions occurring in various conditions of the leaching process. It is an important addition to the knowledge of the mechanisms of this process.

- As an introduction to section 3.3, it should be clearly indicated why the analyses presented in section 3.3 were performed. Why were the results presented for selected parameters and what are they supposed to show, i.e. what is their purpose?

- missing image (k) in Fig.16

- in the title of Table 5, the unit of quantitative content of elements should be given

The discussion of the results is exhaustive and the developed conclusions are correct and supported by the research results.

In my opinion, after making improvements to the manuscript according to the instructions, it will be possible to publish it as valuable scientific material,

kind regards,

Reviewer

Reviewer 2 Report

Comments and Suggestions for Authors

Authors proposed a paper entitled: “The Dissolution Behavior of Pyrite and Chalcopyrite in Their 2 Mixture During Low-Temperature Pressure Oxidation: Kinetic Analysis” for the publication in Materials, mdpi.

The paper has a good scientific impact, but it requires some revisions before publication. For example, the abstract should be revised. A possible solution could be to rephrase some complex concepts; sentences should be streamlined into simpler, more concise forms for better readability. Moreover, redundancies, particularly regarding repeated points, should be eliminated, together with a clarification of some concepts, in order to enhance overall comprehensibility and flow.

Line 36. “In hydrometallurgical processes, the current trend is to perform processes at low 36 temperatures” please add references to support this affirmation and report also specific temperatures and trends of efficiency.

Line 46. “These methods include mechanical activation, the use of surfactants, 46 and the addition of catalysts” authors should be more specific in this affirmation. A too generic definition without specifying surfactants and catalysts types can create confusion in the reader that is not expert of hydrometallurgical methods.

Line 59. “The authors of Refs” I would refer to the names of the authors, not using “refs”.

Line 65. “CuFeS2+4Ag+=2Ag2S+Cu2++Fe2+” the chemical reaction should be reported alone at the center of the subsequent line, and the reference should be reported as “as indicated in Eq. 1 [14]”. Please also check the guidelines of this journal.

Line 74. “At atmospheric pressure,” please refer also to the temperature.

According to my reading, the main problem addressed in this study is the challenge of elemental sulfur passivation during the low-temperature pressure oxidation of sulfide minerals, specifically pyrite and chalcopyrite. This passivating layer significantly hinders reaction rates by restricting reagent diffusion to the reaction surface, which limits the efficiency of leaching processes. Existing methods, while effective to some extent, often lack clarity in the mechanisms of catalytic interactions and the behavior of these minerals under elevated pressure and low-temperature conditions. This research focuses on overcoming these limitations by analyzing the dissolution kinetics of pyrite and chalcopyrite, modeling their oxidation processes, and investigating the role of electrochemical interactions in enhancing their reactivity within a mixed-mineral system. Could you clearly express these concepts while expressing the aims and goals of this paper, at the end of the introduction section? Also in case the authors would like to express a different concept for goals, the advice is to define them exhaustively and clearly, at this point.

Line 124. “Analysis” should be “(sample) characterizations”.

Line 141. Please add CAS numbers.

Line 176. This should become Eq. 2, according to this journal guidelines.

Figure 4. the focus here must be improved, especially in the written part of the legend text.

Results section is characterized by excessive data, equations; therefore, it appears hard to find a unique path to the goals declared above. Authors could summarize key data points in a table and limit detailed discussions to the most critical observations.

Moreover, it seems that the text does not sufficiently highlight the most critical discoveries, leading to a lack of emphasis. An advice could be to end each subsection with a brief summary of the most significant findings and their relevance.

Captions of figures could be more concise, emphasizing their purpose and insights.

The conclusions section, while summarizing key findings, suffers from overgeneralization and redundancy, as it reiterates results without offering deeper interpretations or highlighting their broader implications. The explanation of the interaction between chalcopyrite and pyrite lacks clarity, failing to provide a detailed mechanistic insight. Additionally, the section omits important elements like acknowledging study limitations and suggesting directions for future research, which would add depth and context. Conclusions should focus on synthesizing the study’s broader significance, offering clear mechanistic explanations, avoiding repetitive details, and including a brief discussion of limitations and future prospects.

Reviewer 3 Report

Comments and Suggestions for Authors

Overall, the paper presents a clear research. It includes the key experimental conditions, findings, and insights, making it suitable for an academic audience. The logical flow from the introduction of experimental conditions, the development of mathematical models, kinetic analysis, to the conclusion is well-maintained. The primary findings are highlighted, such as the observation of intra-diffusion limitations in the oxidation process, the changes in the activation energy, and the positive effects of chalcopyrite on pyrite oxidation.

  1. study is focused and presents relevant details regarding the experimental setup (temperature, pressure, leaching) and the essential results, such as the change in activation energy and the influence of chalcopyrite on pyrite oxidation.
  2. Kinetic Analysis: The description of the kinetic analysis is clear, emphasizing the role of diffusion limitations and the passivation of pyrite and chalcopyrite by an elemental sulfur layer.
  3. Experimental Evidence: SEM and EDX mappings provides experimental evidence to support the findings and offers insight into the mechanisms at play, particularly regarding the sulfur distribution on the minerals.
  4. accept

Areas for Improvement:

  1. Specificity in Results:

    • The increase in the oxidation degree of pyrite (54.5% to 80.3%) over 0-230 minutes is quite broad. It would be helpful to briefly explain the significance of this increase in the context of the leaching process.
    • While the increase in activation energy values is mentioned, it would be more impactful to briefly describe how these findings compare to prior research or existing theories in the field.
  2. Terminology:

    • The phrase "diffusion of reagents through the solid reaction product layer" is somewhat vague. Clarifying what "solid reaction product layer" specifically refers to (e.g., an elemental sulfur layer) could enhance clarity.
  3. Impact of Copper and Iron Ions:

    • The claim that the increase in pyrite dissolution is not associated with an increase in copper (II) and iron (III) ions should be explored further. The paper mentions their effect as “insignificant” but does not elaborate on why this is the case, despite the likely expectation that these ions would play a key role. A brief explanation or clarification would improve the completeness of the findings.
  4. Language Precision:

    • The phrase "oxidation mechanism of chalcopyrite and pyrite in their mixture has changed" is a bit general. A more specific description of the nature of this change could help readers better understand the significance of the findings. Is it related to changes in the sulfur layer, or the chemical pathways involved?
  5. Recommendation:

Accept as is, with minor revisions for clarity and depth in a few areas, such as the explanation of ion concentrations and the precise description of the oxidation mechanism changes.

Round 2

Reviewer 2 Report

Comments and Suggestions for Authors

Authors have revised their manuscript according to my issues; however, they declared that they added new references, but they did not. Please add new references where requested.

Author Response

Comment 1. Authors have revised their manuscript according to my issues; however, they declared that they added new references, but they did not. Please add new references where requested.

Answer 1. We apologise to the reviewer. Apparently there was a system failure and the final version of our article with the comments taken into consideration was not uploaded. Please take the time to look at the final version of the manuscript.

Round 3

Reviewer 2 Report

Comments and Suggestions for Authors

In table 5, decimal digits should be separated by points, according to the international convention.

Line 51. "E.g" should be "For example,"

Only one newly reference has been added.

Author Response

Comments 1. In table 5, decimal digits should be separated by points, according to the international convention. 

Responce 1. It has been corrected.

Comments 2. Line 51. "E.g" should be "For example,"

Responce 2. This has been corrected in all cases through the text.

Comments 3. Only one newly reference has been added.

Responce 3. We added three more references.